# Altered White Matter Network Topology in Panic Disorder

**DOI:** 10.3390/jpm13020227

**Published:** 2023-01-27

**Authors:** Molin Jiang, Ping Zhang, Xiangyun Yang, Aihong Yu, Jie Zhang, Xiaoyu Xu, Zhanjiang Li

**Affiliations:** 1Department of Psychosomatic Medicine, Beijing Hospital of Traditional Chinese Medicine, Capital Medical University, Beijing 100010, China; 2The National Clinical Research Center for Mental Disorders, Beijing Key Laboratory of Mental Disorders, Beijing Anding Hospital, Capital Medical University, Beijing 100088, China; 3Chinese Institute for Brain Research, Beijing 102206, China

**Keywords:** panic disorder, white matter, network topology

## Abstract

Panic disorder (PD) is an anxiety disorder that impairs life quality and social function and is associated with distributed brain regions. However, the alteration of the structural network remains unclear in PD patients. This study explored the specific characteristics of the structural brain network in patients with PD by graph theory analysis of diffusion tensor images (DTI). A total of 81 PD patients and 48 matched healthy controls were recruited for this study. The structural networks were constructed, and the network topological properties for individuals were estimated. At the global level, the network efficiency was higher, while the shortest path length and clustering coefficient were lower in the PD group compared to the healthy control (HC) group. At the nodal level, the PD group showed a widespread higher nodal efficiency and lower average shortest path length in the prefrontal, sensorimotor, limbic, insula, and cerebellum regions. Overall, the current results showed that the alteration of information processing in the fear network might play a role in the pathophysiology of PD.

## 1. Introduction

Panic disorder (PD) is characterized by recurrent panic attacks with physical and affective symptoms that appear abruptly and reach peak intensity within minutes. These attacks are usually followed by anticipatory anxiety. The 12-month prevalence rate of PD is 0.3%, and the lifetime prevalence rate is about 0.5% [1]. PD is related to high levels of social, occupational, and physical disability and a high economic cost. 

The traditional pathophysiology of PD originated from Gorman’s hypothesis of the “fear network model (FNM)”, which includes frontal and limbic areas [2]. The dysregulation in the feedback mechanism of the frontal lobe over the fear response of the limbic system may provoke panic attacks. In addition to the frontal and limbic areas, more extended areas have been discovered in recent studies. Two reviews proposed an advanced FNM with sensory and motor regions of the temporo-occipito-parietal cortex and cerebellum based on imaging studies [3,4].

Grey matter (GM) morphological analysis showed alterations of brain regions in FNM, such as significant grey matter (GM) volume reductions in the prefrontal and temporal-parietal cortices [5,6,7], insula [7,8], thalamus [5,6], brainstem [6], and cerebellum [5]. Some other GM morphological studies showed an increased volume of the insula [9] and brainstem [10]. The diffusion tensor imaging (DTI) studies also showed inconsistent results in the alteration of white matter (WM) tracks connecting the GM regions in FNM. Han et al. [11] demonstrated increased WM connectivity of the cingulate. However, other DTI studies revealed decreased WM connectivity of the cingulate [12], fronto-occipital fasciculus [13,14], corpus callosum [13,15], superior longitudinal fasciculus [13,14], and corona radiata [12,15,16]. These findings reflected the structural disruption of the GM and WM in the fear network in PD. The pathophysiology of PD has not been elucidated clearly due to the altered local brain regions in PD patients. Hence, Konishi et al. [12] proposed that the pathophysiology of PD is related to the abnormality in the brain network rather than that in a local brain region. However, only a few studies have assessed the brain network of PD. Cui et al. [17] found an increased functional connection between the somatosensory cortex and the thalamus in PD patients. Another independent component analysis study showed decreased functional connections in the sensorimotor network, default mode network, and cerebellar network [18]. The differential results may be related to different methods. Thus, additional studies are required to further clarify the alteration in the brain network in PD patients.

The topological properties of the human brain’s anatomical networks can be analyzed quantitatively using graph theory [19,20], which is a branch of mathematics and permeates all scientific disciplines. The graph-theoretical approaches are centrally important to understanding the architecture, development, and evolution of brain networks [21] and are widely used in the study of psychiatric disorders such as schizophrenia [20] and depression [22]. Most of the previous structural neuroimaging studies of PD applied traditional voxel-based analyses [12,13,15,23,24,25,26], which could not find imbalanced interactions between the brain regions or extensive pathological changes on a large-scale level. Moreover, the inconsistent results of those structural analyses were insufficient to explain the pathophysiology of PD. The human brain is a complex network of interconnected areas. A few functional magnetic resonance imaging (fMRI) studies revealed abnormal functional connectivity between brain regions in PD patients [17,18]. But analyses examining the whole brain network of PD are limited. As a complex network analysis method, graph theory may detect the altered global connection state of the brain as well as the altered local connection state between brain regions, providing insight into the abnormal structural network in PD patients. The major graph theory parameters include global efficiency, shortest path length, clustering coefficient, and small-world coefficient. The global efficiency, a measure of integration, quantifies a network’s parallel information processing efficiency in a network and is proportional to inverse pathlength [27]. The clustering coefficient measures the ability of segregation and quantifies how well a local graph exchanges information [28].

The current study aimed to look into the topology of structural networks in PD patients. We hypothesized that structural brain network connectivity was altered globally and among brain regions in FNM, especially in PD patients. First, we constructed a structural brain network in PD patients, with brain regions as nodes and WM fiber bundles as edges. Then, we compared the graph theory parameters at both global and nodal levels between PD patients and healthy controls (HC) to verify our hypothesis and provide evidence for the psychopathology of PD.

## 2. Subjects and Methods

### 2.1. Participants

A total of 129 participants were recruited from Beijing Anding Hospital and Beijing Hospital of Traditional Chinese Medicine between March 2020 and June 2022. All participants were screened using the Mini-International Neuropsychiatric Interview. A total of 81 patients were diagnosed with PD conforming to the Diagnostic and Statistical Manual of Mental Disorders (DSM-5) criteria and assessed with the Panic Disorder Severity Scale—Chinese Version (PDSS-CV) and 17-item Hamilton Anxiety Rating Scale (HAMD-17) on the same day. MRI scans were obtained at the Beijing Anding Hospital within 7 days of diagnosis. Then, 48 age-, gender-, and education-matched HCs were recruited from the Anding Hospital public advertisements according to the inclusion and exclusion criteria. 

For participants in the PD group, the inclusion criteria were as follows: (1) diagnosis of PD by a senior psychiatrist conforming to DSM-5; (2) age 18–60 years; (3) PDSS-CV total score > 7 and HAMD-17 total score < 18; (4) with ≥6 years of education; (5) being right-handed; (6) signing the informed consent. The exclusion criteria were as follows: (1) any major physical diseases, such as neurological illnesses, cancer, diabetes, and cardiovascular disease; (2) any infectious diseases; (3) any comorbid mental illnesses such as depression, generalized anxiety disorder, agoraphobia, social phobia, bipolar disorder, obsessive-compulsive disorder, schizophrenia, alcohol addiction, or eating disorders; (4) a history of psychotherapy, physical therapy, or medication within 6 months of enrollment; (5) any contraindication to MRI; and (6) pregnant or lactating women.

The inclusion criteria for the HCs were as follows: (1) age 18–60 years; (2) with ≥6 years of education; (3) right-handed; and (4) age, gender, and years of education matched with the PD group. The exclusion criteria were as follows: (1) any mental illness; (2) any major physical diseases, such as neurological illnesses, cancer, diabetes, and cardiovascular disease; (3) any infectious diseases; (4) women during pregnancy and lactation; (5) any contraindication to MRI.

### 2.2. MRI Acquisition

We acquired DTI and T1-weighted images (T1WI) using a 3T Siemens scanner (Siemens Medical, Erlangen, Germany) at Brain Imaging. During the scanning process, foam pads and ear plugs were used for each participant to reduce head motion and noise.

The DTI data were acquired with an echo-planar imaging sequence using the following parameters: 60 diffusion directions at b = 1000 mm/s^2^ and b = 0 mm/s^2^, repetition time (TR) = 9200 ms, ehco time (TE) = 78 ms, 75 slices, slice thickness = 2 mm, slice spacing = 1.9 mm, field of view = 224 × 224 mm^2^, matrix size = 112 × 112 mm^2^, voxel size = 2 × 2 × 2 mm^3^, and scan time = 690 s.

T1WI were acquired using the three-dimensional (3D) magnetization-prepared rapid gradient-echo sequence with the following parameters: TR = 2530 ms, TE = 1.85 ms, inversion time (TI) = 800 ms, flip angle = 15°, FOV = 256 mm × 256 mm, slices number = 192, thickness = 1.00 mm, no gap, and voxel size = 1.00 mm × 1.00 mm × 1.00 mm.

### 2.3. DTI Preprocessing and Tractography

The raw DTI data were preprocessed using the pipeline for analyzing brain diffusion images (PANDA) software [29] based on FSL (fsl.fmrib.ox.ac.uk). First, non-brain tissues were removed from the raw images, and the brain masks were obtained. Second, the correction was performed for the eddy current distortions and head motion. Third, the tensor model was fitted, and the fractional anisotropy (FA) was calculated. Following the above steps, deterministic tractography was performed with a diffusion toolkit (http://trackvis.org, accessed on 17 December 2022) [30] to produce the connectome reconstruction with the following parameters: fiber assignment by continuous tracking (FACT) algorithms; ﻿minimum FA threshold, 0.2; step length, 0.5 mm; maximum angle threshold, 35°. 

### 2.4. Structural Preprocessing

T1WIs were first segmented into GM, WM, and cerebrospinal fluid. Then, the GM was normalized to Montreal Neurological Institute (MNI) 152 stereotactic space (1 mm^3^ isotropic) with linear and nonlinear registrations. The cortical and subcortical GM was parcellated into 116 brain regions according to the automated anatomical labeling (AAL) atlas [31]. Finally, parcellated T1WIs were aligned to native DTI space to constrain the fiber tracing.

### 2.5. Network Node and Edge Definitions

The nodes of the structural networks were defined as the 116 brain regions according to the AAL atlas [31]. Edge weights were defined as the streamlined numbers connecting each pair of nodes end-to-end. For each subject, a 116 × 116 symmetric weighted network was constructed. The isolated nodes that did not connect to any other nodes (degree of node *i* equals 0) were speculated, but no isolated node was identified in the individual networks. 

### 2.6. Graph Theory Analysis

For the weighted structural networks, both the global and nodal network metrics were calculated. The global metrics included shortest path (*S_p_*), global efficiency (*E_glob_*), clustering coefficient (*C_p_*), and small-world coefficient (*sigma*) [32], while the nodal metrics included nodal efficiency, nodal shortest path, and nodal clustering coefficient. The definitions and calculation formula of all the metrics for the whole graph and nodes are introduced below.

For weighted networks, the length of each edge is the reciprocal of the edge weight, *1/w_ij_*. The “shortest path” refers to the path connecting the two nodes with the fewest edges. The shortest path length (*L_ij_*) between a pair of nodes (*i* and *j*) is defined as the sum of the path lengths in the shortest path. The measure of global *S_p_* is calculated by averaging the *L_ij_* of all pairs of nodes in the network.

The inverse *L_ij_* is related to the efficiency. For a network *G* with *N* nodes, the *E_glob_* is the average efficiency over all pairs of nodes and estimates the global efficiency of the parallel information transfer in the network [33]. It was calculated using the following formula
$$E_{glob}\left(G\right)=\frac{1}{N\left(N-1\right)}\underset{i\neq j}{\sum }\frac{1}{L_{ij}}$$

The efficiency between node *i* and all the other nodes in the network was calculated [34] using the following formula
$$E_{nodal\_i}=\frac{1}{N-1}\underset{i\neq j}{\sum }\frac{1}{L_{ij}}$$

*C_p_* is a measure of how nodes in a network tend to cluster together. The upper formula was used for nodal clustering coefficient, while the lower was for global *C_p_*. *k_i_* is the degree of node i, and $\hat{w}$ is the scaled weight based on the mean of all weights [35]. The global *C_p_* is the average nodal clustering coefficient.
$$C_{p}\left(i\right)=\frac{2}{k_{i}\left(k_{i}-1\right)}\underset{j,\;k}{\sum }{\left(\hat{w_{ij}}\;\hat{w_{ik}\;}\hat{w_{jk}}\right)}^{1/3}$$
$$C_{p}=\frac{1}{N}\underset{i}{\sum }C_{p}\left(i\right)$$

A small-world network is characterized by a higher clustering coefficient than random networks and a similar shortest path length to random networks. The small-network coefficient sigma can be calculated using the formula below [36]. A small-world network is a network with a sigma larger than one.
$$sigma=\frac{C_{p-real}/C_{p-random}}{S_{p-real}/S_{p-random}}$$

GRETNA was used to calculate the graph theory metrics above [37], and BrainNet Viewer was used to visually represent the edges and nodes in the brain [38]. These analyses, including preprocessing and graph theory analysis, were simplified through a cloud platform (http://www.humanbrain.cn, accessed on 23 September 2022, Beijing Intelligent Brain Cloud, Inc.). The entire analysis is presented in a flowchart (Figure 1).

### 2.7. Statistical Analysis

Demographic and behavioral data were evaluated in SPSS v. 20.0 (IBM, Armonk, NY, USA). The 2-tailed test was used for continuous variables with a normal distribution. The Mann–Whitney test was performed for non-normally distributed variables. The χ2 test was used to compare categorical variables. The graph metrics were compared between the two groups controlling gender and age, and the significance was tested through a nonparametric permutation test with 10,000 permutations. The tests for global and nodal metrics were corrected for multiple comparisons through false discovery rate (FDR) correction. As 4 global measures and 116 nodes were compared, we corrected the *p* values for 4 and 116 times in global and nodal level analysis separately. The functions of the permutation test and FDR correction were provided by Gretna in MATLAB. 

To evaluate the ability of the topological differences to classify PD, we first estimated 4 logistic models by defining group as the binomial dependent values and global topological measures as independent values. Gender and age were controlling. Then, the receiver operating characteristic (ROC) curves for global metrics were drawn, and the area under the curves was calculated using the pROC package. The permutation test was applied to determine whether the area under the ROC curve (AUROC) was significantly higher than the values expected by chance. Specifically, we permuted the group labels (PD or HC) across the entire sample 1000 times without replacement, and the model estimation and ROC construction were reapplied each time. The *P* value was calculated by dividing the number of permutations that showed a higher value than the actual value for the real sample by the total permutation times (1000). These analyses were performed using R 4.1.0 [39].

The correlations between topological properties and the PDSS-CV score were evaluated by Pearson’s correlation in SPSS version 20.0.

## 3. Results

### 3.1. Demographics

The participant demographic characteristics are compared in Table 1. The sample consisted of 81 PDs and 48 HCs. All participants were right-handed and of Han Chinese descent. No significant difference was observed between groups with respect to age (*t* = −1.561, *p* = 0.115), gender (*χ2* = 0.597, *p* = 0.562), and educational years (*t* = 6.606, *p* = 0.546). The mean PDSS-CV score of the PD patients was 12.28 ± 2.68. The HAMD-17 score of the PD patients was 9 (6–12), while that of the HCs was 0 (0-0) (*z* = −9.606, *p* < 0.001). 

### 3.2. Global and Nodal Network Changes 

Patients with PD showed increased global efficiency compared to HCs (18.10 ± 1.96 vs. 17.08 ± 2.07; *t* = 2.754 FDR corrected, *p* = 0.005) (Figure 2A). The nodes with significantly higher global efficiency were distributed widely across the bilateral frontal, parietal, temporal, and occipital lobes; the right insula; the right thalamus; the bilateral limbic region; and the right cerebellum (FDR corrected *p* < 0.05) (Figure 2B). The node names and the mean values of each node are displayed in Table 2. The AUROC of global efficiency was 0.663 (Figure 2C), and the AUROC of global efficiency was significantly higher than random models (*p* = 0.017).

For the clustering coefficient, patients with PD showed a significantly lower global clustering coefficient than HCs (0.019 ± 0.005 vs. 0.020 ± 0.005, *t* = –1.857 FDR corrected, *p* = 0.044) (Figure 3A). The AUROC of the clustering coefficient is 0.624 (Figure 3C), and the AUROC of the global clustering coefficient was significantly higher than random models (*p* = 0.021). No significant difference was detected in the nodal clustering coefficients between the two groups.

The comparison demonstrated a significantly decreased global shortest path length in patients with PD (0.055 ± 0.006 vs. 0.059 ± 0.007, *t* = −3.034 FDR corrected, *p* = 0.005) (Figure 3A). The nodes contributed to the shorter path length distributed across the bilateral frontal, parietal, temporal, and occipital lobes; the right insula; the right thalamus; the bilateral limbic region; and the right cerebellum (FDR corrected *p* < 0.05) (Figure 3B). The AAL regions with significant differences in the nodal shortest path length are listed in Table 3. The AUROC of the shortest path was 0.667 (Figure 3C), and the AUROC of the global shortest path length was significantly higher than random models (*p* = 0.019).

For sigma, no significant difference was found between the PD and HC groups before FDR correction (2.864 ± 0.182 vs. 2.858 ± 0.196, *t* = 0.273 *p* = 0.387)

And no significant correlations were observed between the graph theory metrics and PDSS-CV scores, HAMD-17 scores, or PD disease course (*p* > 0.05 before FDR correction).

## 4. Discussion

In the present study, we constructed WM connectivity networks using DTI imaging in PD patients and HCs and found widespread differences in the structural network topological properties between the two groups. These findings provided a detailed look at the structural connectome for panic disorder psychopathology. 

A higher global efficiency, a lower global shortest path, and a lower clustering coefficient were found in patients with PD than in HCs, though no significant difference was observed in the comparison of *sigma* between the two groups. These findings might indicate a potential alteration in the “small world” feature in the brain network of PD patients. The “small world” network has a shortest path similar to a random network and a high clustering coefficient similar to a regular network [40]. As a result, the “small world” network has a high communication speed while using little energy. Among these topological indicators, the shortest path quantifies the ability to propagate parallel information. Global efficiency is proportional to the inverse shortest path length and reflects the functional integration of a network [41]. The clustering coefficient measures the cliquishness of a typical neighborhood and reflects the functional segregation and fault tolerance of a network [33,40,42]. A healthy brain achieves a balance between global integration and local segregation. In this study, PD patients showed a decreased clustering coefficient, which suggested low functional segregation and increased global efficiency, further indicating high functional integration. This imbalance reflected a tendency to form random networks with high energy costs and low fault tolerance in the brain networks of PD patients. The AUROC of global efficiency, shortest path, and clustering coefficient was between 0.6 and 0.7, indicating moderate classification accuracy. The network metrics of global efficiency, shortest path, and clustering coefficient could discriminate PD patients from HC, although additional studies are required. 

Interestingly, a high nodal efficiency and low nodal shortest path were observed in several brain regions, including the temporal-parietal-occipital region, the insula, the thalamus, the limbic region, and the cerebellum, in PD patients compared to HCs. According to the traditional FNW proposed by Gorman et al. [2], the frontal lobe was dysregulated in the feedback mechanism for cognitive control over the primitive response of the limbic system in PD patients. The exaggerated reaction of the limbic system induced panic attacks. However, more brain regions were identified, and an advanced FNW was proposed by Lai et al. [3] and Zhang et al. [4]. The insula integrated the filtered sensory information via the thalamus from the visuospatial and other sensory modalities related to the occipital, parietal, and temporal lobes. To control the fear response, the insula collaborated with the prefrontal lobe. The excessive sensory information detected by the temporal, parietal, and occipital lobes, the dysregulated integrative function of the insula, the abnormal feedback mechanism of the frontal lobe, and the exaggerated responses of the limbic system together led to panic attacks.

This study showed higher nodal efficiency and a lower nodal shortest path in almost all parts of the FNM, probably indicating abnormal information processing in the fear network of PD patients. The temporal-parietal-occipital region is related to sensory-related function, including visuospatial attention [43,44,45], auditory-spatial attention [46], somatosensory procession [47], and fear stimuli procession [44], that works as an entrance for sensory information into FNM. The disruption of the temporal-parietal-occipital region has been widely reported in many studies focused on PD [5,9,48,49,50]. Increased nodal efficiency in the temporal-parietal-occipital region might indicate that excessive and exaggerated sensory information, especially the fear signal, is processed and input into the FNM.

In FNM, the thalamus is the brain region that filters and integrates information from the temporal-parietal-occipital region. Previous functional network analyses showed increased functional connectivity between the thalamus and the precentral and postcentral gyrus in patients with PD [17]. This WM network study showed elevated nodal efficiency with a reduced nodal shortest path in the thalamus of PD patients. These findings might reflect correspondingly increased information processing in the thalamus to deal with the biased information from the temporal-parietal-occipital region. 

Another critical part of FNM is the insula, which integrates information from the thalamus and controls the fear response with the frontal lobe [3]. In previous fMRI studies, fear of cardiovascular symptoms [51], response to visual threat [52], and anxiety sensitivity during emotional face processing [53,54] were related to the activation of the insula in PD patients. The insula is a crucial area for the somatic and cognitive pathophysiology of PD. In the present study, we observed increased nodal efficiency in the insula, which might indicate enhanced information processing in the insula to regulate cognition and control fear responses. 

The orbitofrontal cortex (OFC), located in the prefrontal lobe, is the emotion-regulating brain region [55]. The role of OFC in cognition involves encoding conflicting information and subsequent adjustment of executive control [56]. Lai et al. [57] demonstrated that the dysfunction of the network composed of the OFG, inferior frontal gyrus, and superior temporal gyrus was related to PD symptoms. This study showed higher nodal efficiency and a lower nodal shortest path in the OFC region of PD patients, indicating elevated information processing in the OFC to integrate cognitive information and feedback to inhibit the primary fear response of the limbic system. However, the limbic system was not successfully inhibited in PD, as nodal efficiency in the limbic region increased in this study. This phenomenon suggested that the limbic region may be overactive in response to thalamic stimuli and is rarely suppressed by the insula and frontal lobe, resulting in a fear response and panic attacks.

Moreover, this study showed increased nodal efficiency in the precentral gyrus and supplementary motor area. Lai et al. suggested that the precentral gyrus of PD patients was a network hub in a whole brain fMRI study [58]. It strengthened the potential background of the precentral gyrus for panic attack-related motor symptoms, such as fright, fear of losing control, and escape behavior. 

The cerebellum is a critical structure related to PD in this study. Traditionally, the main function of the cerebellum has been to coordinate movement. However, MRI studies showed that the cerebellum could be activated by non-motor tasks [59,60] as it is a vital part of the autonomic neural network [61]. Autonomic nervous system disturbances, such as palpitations and blood pressure changes, are common in PD patients. The cerebellum vermis is associated with the fear response and plays a crucial role in fear conditioning and maintenance [62]. An independent component analysis study revealed abnormal functional connectivity in the cerebellar network in PD patients [18]. The present study showed a widely increased nodal efficiency and the shortest nodal path in the cerebellum, which might indicate biased information processing in the cerebellum of PD patients. 

No correlation was observed between the network topological properties and PDSS-CV scores. The negative result might be due to the relatively mild severity and short disease course of the patients enrolled in the current study. About half of the PD patients had disease progression within 6 months, and the alteration of structural network properties was slight. It might be hard to detect a significant correlation between topological attributes and PDSS scores when topological alteration is slight and behavioral severity is mild to moderate. In previous studies, other anxiety measurement scores, such as the Hamilton Anxiety Scale (HAMA), Beck Anxiety Inventory (BAI), and Anxiety Sensitivity Index (ASI-R), were found to be related to topological properties and white matter tracks based on f-MRI [50] and DTI data [13,15,63]. However, in the current study, no anxiety measurements except the PDSS were applied. So more attention should be paid to severe PD patients with long disease courses, and more anxiety measurement-related psychiatric evaluations should be applied to further explore the correlation between topological alteration and clinical symptoms in the future.

## 5. Limitation

Nevertheless, the present study has limitations. It was a cross-sectional design. Thus, longitudinal exploration is required to confirm our findings. Besides, most PD patients in the current study were mild or moderate, as serious PD patients often fail to complete the MRI scan. Therefore, this sample might not represent the general brain imaging characteristics of patients with PD. 

## 6. Conclusions

In this study, we investigated the alterations in the structural network topology in patients with PD using graph theory analysis. The WM brain network of PD patients showed a potential tendency toward a random network with a high energy cost and low fault tolerance. The efficiency of information transmission was higher in almost all parts of the FNM and in the motor region and cerebellum in patients with PD. The alteration of information processing in the fear network might play a crucial role in the pathophysiology of PD. 

## Figures and Tables

**Figure 1 jpm-13-00227-f001:**
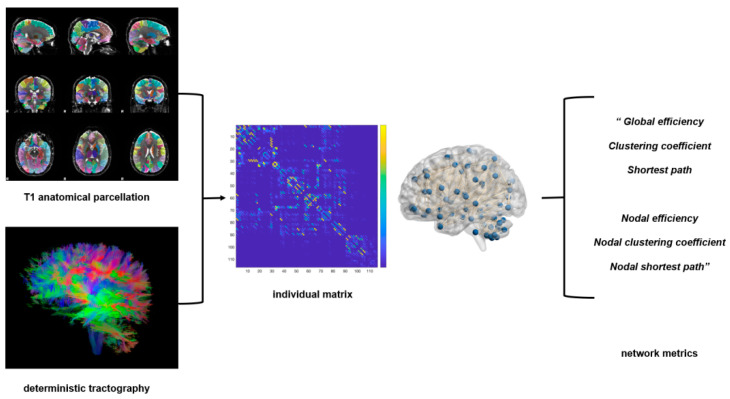
Flow chart of structural network construction. Deterministic tractography was performed for connectome reconstruction, while the cerebral cortex was parcellated into 116 brain regions according to the automated anatomical labeling (AAL) atlas and then aligned to native DTI space to constrain the fiber tracing. The nodes of the structural networks were defined as the 116 brain regions according to the AAL atlas. Edge weights were defined as the streamline counts connecting each pair of nodes end-to-end. Then, both the global and nodal network metrics were calculated. The global metrics included global efficiency, shortest path, and clustering coefficient; the nodal metrics included nodal efficiency, nodal shortest path, and nodal clustering coefficient.

**Figure 2 jpm-13-00227-f002:**
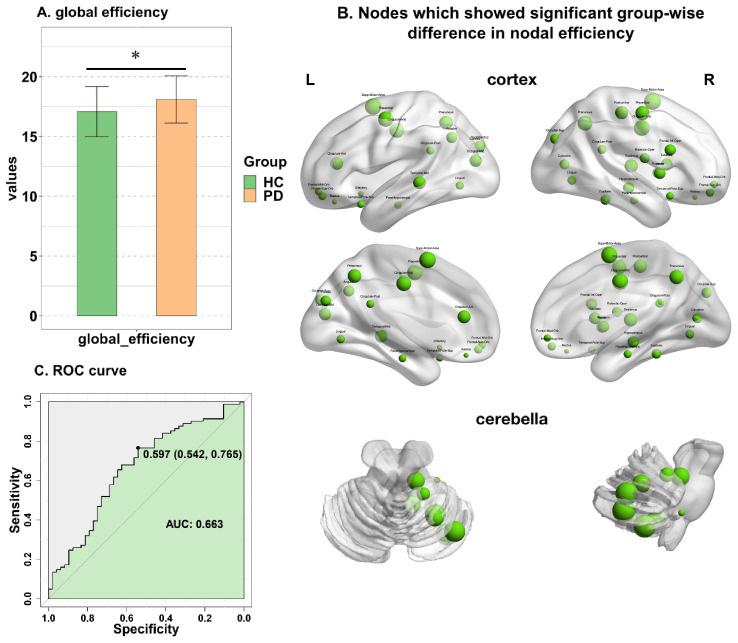
Differences in global and nodal efficiencies between HC and PD groups. (**A**) * showed s ignificant differences in global efficiency between HC and PD groups. (**B**) Nodes, which showed a significant group-wise difference in nodal efficiency. The size of nodes refers to the magnitude of nodal efficiency. (**C**) ROC curve for the logistic model of global efficiency, the optimal threshold leads to a specificity of 0.542 and a sensitivity of 0.765.

**Figure 3 jpm-13-00227-f003:**
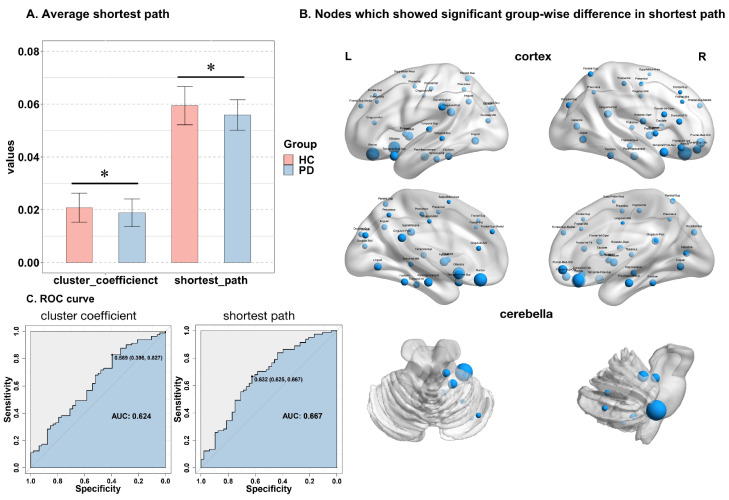
Global cluster coefficient, global average shortest path, and nodal shortest path length. (**A**) * showed significant differences in average cluster coefficient and shortest path between HC and PD groups. (**B**) Nodes, which showed a significant group-wise difference in the shortest path. The size of nodes refers to the magnitude of the nodal shortest path. (**C**) ROC curves for the logistic models of global cluster coefficient and shortest path. The optimal threshold of the cluster coefficient leads to a specificity of 0.396 and a sensitivity of 0.827, while the optimal threshold of the shortest path leads to a specificity of 0.625 and a sensitivity of 0.667.

**Table 1 jpm-13-00227-t001:** Demographics.

	Patients with PD (N = 81)	Healthy Controls (N = 48)	t/z/χ2	*p*-Value
Gender (%)	M29 (35.8)	M14 (29.2)	0.336	0.562
	F52(64.2)	F34 (70.8)		
Age, mean (SD), years-old	34.09 (9.55)	36.88 (9.82)	−1.561	0.115
Educational years, mean (SD)	15.77 (2.33)	15.50 (2.54)	6.606	0.546
Duration of illness, P_50_ (P_25_, P_75_), months	12 (4.34)			
PDSS, mean (SD)	12.28 (2.68)	-		-
HAMD, P_50_ (P_25_, P_75_)	9 (6.12)	0 (0.0)	−9.606	0.000

SD: standard deviation.

**Table 2 jpm-13-00227-t002:** Nodal efficiency.

Nodal Name	Patients with PD	Healthy Controls	*t*-Value	*p*-Value
	Mean (SD)	Mean (SD)		
Precentral_L	30.858 (4.425)	28.657 (5.104)	2.633	0.029
Precentral_R	28.415 (4.427)	26.696 (5.104)	2.079	0.049
Frontal_Sup_Orb_L	17.162 (4.548)	15.243 (4.571)	2.393	0.039
Frontal_Sup_Orb_R	17.362 (3.861)	15.888 (3.801)	2.173	0.041
Frontal_Mid_Orb_L	15.666 (4.850)	13.498 (3.569)	1.997	0.029
Frontal_Inf_Oper_R	22.774 (3.320)	21.311 (3.758)	2.733	0.039
Rolandic_Oper_R	20.464 (2.623)	19.275 (2.593)	2.311	0.036
Supp_Motor_Area_L	33.250 (5.238)	30.536 (4.993)	2.417	0.029
Supp_Motor_Area_R	33.771 (4.951)	30.857 (4.482)	2.848	0.029
Olfactory_L	11.543 (2.067)	10.752 (2.155)	3.269	0.041
Frontal_Med_Orb_R	16.001 (3.218)	14.114 (4.182)	2.167	0.029
Rectus_L	11.051 (2.448)	9.985 (2.498)	2.841	0.039
Rectus_R	10.593 (2.488)	9.474 (1.856)	2.337	0.029
Insula_R	23.498 (2.768)	22.028 (2.859)	2.656	0.029
Cingulum_Ant_L	26.086 (3.087)	24.727 (3.383)	2.777	0.039
Cingulum_Mid_L	29.781 (3.766)	28.192 (3.726)	2.288	0.039
Cingulum_Mid_R	31.892 (4.110)	29.999 (3.930)	2.300	0.034
Cingulum_Post_L	16.544 (2.663)	15.539 (2.556)	2.487	0.041
Cingulum_Post_R	18.223 (3.207)	16.859 (3.053)	2.101	0.034
Hippocampus_R	22.943 (2.761)	21.580 (2.725)	2.390	0.029
ParaHippocampal_L	16.138 (2.207)	15.193 (2.090)	2.747	0.034
ParaHippocampal_R	16.622 (2.244)	15.355 (2.160)	2.392	0.029
Calcarine_R	20.348 (3.097)	19.088 (2.729)	3.178	0.039
Cuneus_L	21.585 (3.225)	20.190 (2.977)	2.292	0.039
Lingual_L	17.287 (2.224)	16.198 (1.748)	2.387	0.029
Lingual_R	18.315 (2.503)	17.067 (2.184)	2.940	0.029
Occipital_Sup_L	22.690 (3.234)	21.367 (3.083)	2.950	0.039
Occipital_Sup_R	21.273 (3.432)	19.689 (3.014)	2.225	0.032
Occipital_Mid_L	24.463 (3.563)	23.071 (3.322)	2.581	0.047
Fusiform_R	19.835 (2.292)	18.928 (2.241)	2.116	0.041
Postcentral_R	26.470 (4.304)	24.870 (4.254)	2.183	0.047
Angular_L	22.633 (3.638)	21.054 (3.026)	2.106	0.034
Precuneus_L	27.624 (3.806)	25.806 (3.811)	2.452	0.029
Precuneus_R	30.056 (4.290)	27.721 (4.013)	2.621	0.029
Caudate_R	23.973 (3.596)	22.503 (3.730)	3.009	0.039
Putamen_R	23.226 (3.675)	21.382 (3.433)	2.203	0.029
Thalamus_R	26.984 (4.141)	25.214 (4.357)	2.736	0.034
Temporal_Pole_Sup_L	15.549 (2.470)	14.487 (2.726)	2.348	0.039
Temporal_Pole_Sup_R	15.800 (2.573)	14.655 (2.428)	2.196	0.034
Temporal_Mid_L	25.362 (4.290)	23.607 (3.987)	2.417	0.039
Cerebellum_Crus1_R	9.502 (2.327)	8.578 (1.618)	2.218	0.030
Cerebellum_Crus2_R	9.585 (2.013)	8.707(1.696)	2.558	0.029
Cerebellum_3_R	8.742 (3.121)	7.488 (2.366)	2.681	0.034
Cerebellum_4_5_R	7.420 (1.688)	6.811(1.383)	2.440	0.039
Cerebellum_6_R	8.486 (1.989)	7.819 (1.429)	2.239	0.040
Cerebellum_8_R	9.590 (2.159)	8.592 (1.549)	2.132	0.029
Cerebellum_9_R	8.858 (2.298)	7.848 (1.416)	2.927	0.029
Cerebellum_10_R	5.486 (1.488)	4.859 (1.161)	2.933	0.034

SD: standard deviation; *p*-value is FDR corrected.

**Table 3 jpm-13-00227-t003:** Nodal average shortest path length.

Nodal_Name	Patients with PD	Healthy Controls	*t*-Value	*p*-Value
	Mean (SD)	Mean (SD)		
Precentral_L	0.033 (0.004)	0.036 (0.007)	−3.135	0.020
Precentral_R	0.036 (0.005)	0.039 (0.007)	−2.568	0.031
Frontal_Sup_L	0.031 (0.005)	0.034 (0.007)	−2.204	0.039
Frontal_Sup_R	0.033 (0.005)	0.035 (0.007)	−1.977	0.048
Frontal_Sup_Orb_R	0.061 (0.014)	0.068 (0.023)	−2.294	0.038
Frontal_Mid_L	0.031 (0.004)	0.034 (0.006)	−2.550	0.031
Frontal_Mid_R	0.034 (0.005)	0.037 (0.007)	−2.281	0.038
Frontal_Inf_Oper_R	0.045 (0.006)	0.048 (0.009)	−2.632	0.029
Frontal_Inf_Tri_R	0.047 (0.008)	0.05 (0.01)	−2.195	0.038
Frontal_Inf_Orb_R	0.055 (0.008)	0.059 (0.011)	−2.129	0.041
Rolandic_Oper_R	0.05 (0.006)	0.053 (0.007)	−2.621	0.028
Supp_Motor_Area_L	0.031 (0.005)	0.034 (0.006)	−3.087	0.022
Supp_Motor_Area_R	0.03 (0.004)	0.033 (0.005)	−3.574	0.020
Olfactory_L	0.089 (0.016)	0.097 (0.021)	−2.412	0.033
Frontal_Sup_Medial_L	0.038 (0.005)	0.041 (0.007)	−2.252	0.039
Frontal_Sup_Medial_R	0.041 (0.006)	0.044 (0.009)	−2.211	0.038
Frontal_Med_Orb_R	0.065 (0.015)	0.08 (0.037)	−3.084	0.020
Rectus_L	0.096 (0.028)	0.108 (0.036)	−2.080	0.046
Rectus_R	0.099 (0.021)	0.11 (0.024)	−2.693	0.028
Insula_R	0.043 (0.005)	0.046 (0.006)	−3.118	0.020
Cingulum_Ant_L	0.039 (0.004)	0.041 (0.006)	−2.527	0.028
Cingulum_Mid_L	0.034 (0.004)	0.036 (0.005)	−2.471	0.032
Cingulum_Mid_R	0.032 (0.004)	0.034 (0.004)	−2.657	0.028
Cingulum_Post_L	0.062 (0.01)	0.066 (0.012)	−2.184	0.041
Cingulum_Post_R	0.056 (0.009)	0.061 (0.01)	−2.633	0.028
Hippocampus_R	0.044 (0.005)	0.047 (0.006)	−2.797	0.027
ParaHippocampal_L	0.063 (0.009)	0.067 (0.01)	−2.336	0.038
ParaHippocampal_R	0.061 (0.008)	0.066 (0.009)	−3.294	0.020
Calcarine_R	0.05 (0.008)	0.053 (0.007)	−2.207	0.041
Cuneus_L	0.047 (0.008)	0.051 (0.008)	−2.213	0.039
Lingual_L	0.059 (0.007)	0.062 (0.007)	−2.949	0.022
Lingual_R	0.056 (0.007)	0.06 (0.008)	−3.018	0.020
Occipital_Sup_L	0.045 (0.006)	0.048 (0.007)	−2.306	0.038
Occipital_Sup_R	0.048 (0.007)	0.052 (0.008)	−2.681	0.027
Occipital_Mid_L	0.042 (0.006)	0.044 (0.006)	−2.226	0.038
Fusiform_L	0.055 (0.008)	0.057 (0.007)	−1.935	0.048
Fusiform_R	0.051 (0.006)	0.054 (0.007)	−2.247	0.038
Postcentral_L	0.037 (0.005)	0.04 (0.007)	−2.135	0.041
Postcentral_R	0.039 (0.006)	0.041 (0.007)	−2.344	0.035
Parietal_Sup_L	0.041 (0.007)	0.044 (0.007)	−2.065	0.041
Parietal_Sup_R	0.042 (0.008)	0.045 (0.009)	−2.046	0.047
SupraMarginal_L	0.06 (0.009)	0.064 (0.012)	−2.105	0.041
Angular_L	0.045 (0.007)	0.049 (0.007)	−2.476	0.031
Precuneus_L	0.037 (0.005)	0.04 (0.006)	−2.704	0.028
Precuneus_R	0.034 (0.005)	0.037 (0.005)	−3.165	0.020
Caudate_R	0.043 (0.006)	0.046 (0.007)	−2.412	0.032
Putamen_L	0.046 (0.008)	0.049 (0.008)	−2.036	0.041
Putamen_R	0.044 (0.006)	0.048 (0.007)	−3.181	0.020
Pallidum_L	0.064 (0.01)	0.068 (0.011)	−2.082	0.043
Pallidum_R	0.059 (0.008)	0.062 (0.011)	−1.982	0.048
Thalamus_R	0.038 (0.006)	0.041 (0.007)	−2.674	0.028
Temporal_Sup_L	0.047 (0.007)	0.05 (0.008)	−1.950	0.048
Temporal_Pole_Sup_L	0.066 (0.01)	0.071 (0.013)	−2.616	0.028
Temporal_Pole_Sup_R	0.065 (0.011)	0.07 (0.012)	−2.488	0.032
Temporal_Mid_L	0.04 (0.007)	0.044 (0.007)	−2.373	0.035
Temporal_Inf_L	0.048 (0.007)	0.051 (0.007)	−2.288	0.048
Cerebellum_Crus1_R	0.111 (0.025)	0.121 (0.025)	−2.554	0.038
Cerebellum_Crus2_R	0.109 (0.023)	0.119 (0.025)	−2.253	0.029
Cerebellum_3_R	0.128 (0.041)	0.143 (0.034)	−2.227	0.038
Cerebellum_4_5_R	0.141 (0.03)	0.153 (0.032)	−2.824	0.038
Cerebellum_8_R	0.109 (0.023)	0.12 (0.023)	−2.736	0.027
Cerebellum_9_R	0.119 (0.027)	0.132 (0.025)	−2.136	0.027
Cerebellum_10_R	0.197 (0.06)	0.219 (0.063)	−3.135	0.043

SD: standard deviation; *p*-value is FDR corrected.

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
