# Peer review of "Altered White Matter Network Topology in Panic Disorder"

_jpm, 2023, doi:10.3390/jpm13020227_

Round 1

Reviewer 1 Report

This is an interesting study and the authors have collected a good dataset. The authors performed proper methodologies to analyze the dataset, and the paper is generally well written and structured.  However, I still have several considerations to improve this paper as below.

Major issues:

1. I don't agree with that the slightly differences of topologies between PD patients and healthy controls could deduce to disruption of "fear" network without other evidences, such as behavioral relevance. Besides, the nodal topological alterations are calculated based on the structural connections, they could not reflect the real activations. So it should be very careful to speculate that those regions/networks are "hyper-activation". The increased efficiency in PD patients might related with abnormal modularity or reconfiguration, which also needs more evidences to prove.

2. (Page 5, Last Para) What statistical comparison did you perform? Please specify and provide the corresponding statistical values in results and tables.

3. (Page 6, Para 3) Please specify how did you do the FDR correction for global measures?  And how did you calculate the ROC curve for global measures?

4. (Page 6, Para 4) Regarding to the average and SD, the difference between global clustering coefficient is so weak, how could it pass the FDR correction? Please provide the statistical value.

5. (Page 6, Para 6) In results, authors mentioned they did not observe any significant correlation between PDSS-CV and topological properties. But how did authors implement the correlation analyses? Please provide details.

6. (Page 7, Para 2) The author mentioned the potential alteration of "small world" properties in PD patients. Why not calculate that properties and test?

Minor issues:

1. Some typos need to be corrected.

2. Please give some descriptions of tables, such as what the "P-value" is? Original or already corrected?

3. Please make the values be specified to same decimal precision, for example three decimal places.

4. Please use different color to indicate which nodal regions are increased/decreased in PD patients for Figure 2 and 3. 

5. What are the different sizes of nodes in Figure 2/3 mean? Please give a description. 

Author Response

Dear reviewer:

Thank you very much for the constructive comments and suggestions on our manuscript entitled “Altered White Matter Network Topology in Panic Disorder” (manuscript ID-jpm-2137143). We have studied the comments carefully and have made modification by “track change” in word. The correction and responds to reviewer’s comments is in the attachment file.

Thanks again for your valuable comments and hope that the correction will meet with approval.

Best regards,

Molin Jiang

Reviewer 2 Report

This study did a preliminarily investigate about WM fiber network and its topological features in panic disorder, which is rarely reported to date. I would have some minor comments:

1. In the introduction, "structural" was used to descirbe both GM morphological analysis by T1 and WM fiber measures by DTI. Which is confusion. Please make the statement more clearly.

2. The aothors claimed that functional connectivity is constrained by structural connectivity and structrual connectivity is more stable. However, they did not give enough supporting reference. Moreover, structural connectivity may have its own shortage, etc. less senstitive. Given that the authors did not use both FC and SC, and compared them or showing how SC constrained FC. I would suggest not start this topic in this manuscript.

3. Graph theory have its advantage. However why it is useful for PD studies? What additional information was expected for PD, compared to previous studies?

4. For AUROC, is them significantly larger than radom level?

5. The correlation results between brain and behavior should be reported even it is not significant. Is this "unsignificant" is under multiple comparison correction (given you have many nodes and several graph features).

Author Response

(The authors gave the same response as above.)

Reviewer 3 Report

Ladies and Gentlemen,

below, in points, I present my questions, reservations, suggestions for changes and comments to your manuscript. Please respond to each of the questions.

1. Please expand the abbreviation "HC" in the abstract and in the first part of "INTRODUCTION".

2. Exclusion criteria: how do you define "any major physical (...) diseases". ? I propose to expand on this exclusion criterion because it is not clear.
3. Why the exclusions include: "pregnancy and lactation"? Is this a contraindication to this type of research?
4. The mention of http://trackvis.org requires a citation.
5. Please add a citation to the excerpt: "The nodes of the structural networks were defined as the 116 brain regions according to the AAL atlas [Add reference of AAL atlas]. ".
6. Please correctly quote the passage "(http://www.humanbrain.cn, Beijing Intelligent Brain Cloud, Inc.).".

The study was properly designed. I am missing here some clarification of some issues related to the criteria for inclusion or exclusion of patients in the study.

Ethical issues are not addressed throughout the article.
Have you obtained approval from your local ethics committee to conduct this study?
Was the study recorded somewhere?
Did the patients undergoing the study express a conscious and voluntary will to undergo the study?
How did they consent?
Were patients hospitalized during the study?
Were the patients under psychiatric care?

Author Response

(The authors gave the same response as above.)

Round 2

Reviewer 3 Report

Dear authors.

Thank you for your great commitment to correcting all the comments made. All my doubts have been resolved.